# Surface protein imputation from single cell transcriptomes by deep neural networks

Zilu Zhou[1,2], Chengzhong Ye [3], Jingshu Wang[4] & Nancy R. Zhang [2]*

While single cell RNA sequencing (scRNA-seq) is invaluable for studying cell populations, cell-surface proteins are often integral markers of cellular function and serve as primary targets for therapeutic intervention. Here we propose a transfer learning framework, single cell Transcriptome to Protein prediction with deep neural network (cTP-net), to impute surface protein abundances from scRNA-seq data by learning from existing single-cell multi-omic resources.

[1] Graduate Group in Genomics and Computational Biology, University of Pennsylvania, Philadelphia, PA, USA. [2] Department of Statistics, University of Pennsylvania, Philadelphia, PA, USA. [3] School of Medicine, Tsinghua University, Beijing, China. [4] Department of Statistics, The University of Chicago, Chicago, IL, USA. *email: nzh@wharton.upenn.edu

Recent technological advances allow the simultaneous profiling, across many cells in parallel, of multiple omics features in the same cell[1–5]. In particular, high throughput quantification of the transcriptome and a selected panel of cell surface proteins in the same cell is now feasible through the REAP-seq and CITE-seq protocols[2,3]. Cell surface proteins can serve as integral markers of specific cellular functions and primary targets for therapeutic intervention. Immunophenotyping by cell surface proteins has been an indispensable tool in hematopoiesis, immunology and cancer research during the past 30 years. Yet, due to technological barriers and cost considerations, most single cell studies, including Human Cell Atlas project[6], quantify the transcriptome only and do not have cell-matched measurements of relevant surface proteins[7,8]. Sometimes, which cell types and corresponding surface proteins are essential become apparent only after exploration by scRNA-seq. This motivates our inquiry of whether protein abundances in individual cells can be accurately imputed by the cell's transcriptome.

We propose cTP-net (single cell Transcriptome to Protein prediction with deep neural network), a transfer learning approach based on deep neural networks that imputes surface protein abundances for scRNA-seq data. Through comprehensive benchmark evaluations and applications to Human Cell Atlas and acute myeloid leukemia (AML) data sets, we show that cTP-net outperform existing methods and can transfer information from training data to accurately impute 24 immunophenotype markers, which achieve a more detailed characterization of cellular state and cellular phenotypes than transcriptome measurements alone. cTP-net relies, for model training, on accumulating public data of cells with paired transcriptome and surface protein measurements.

## Results

**Method overview.** An overview of cTP-net is shown in Fig. 1a. Studies based on both CITE-seq and REAP-seq have shown that the relative abundance of most surface proteins, at the level of individual cells, is only weakly correlated with the relative abundance of the RNA of its corresponding gene[2,3,9]. This is due to technical factors such as RNA and protein measurement error[10], as well as inherent stochasticity in RNA processing, translation and protein transport[11–15]. To accurately impute surface protein abundance from scRNA-seq data, cTP-net employs two steps: (1) denoising of the scRNA-seq count matrix and (2) imputation based on the denoised data through a transcriptome-protein mapping (Fig. 1a). The initial denoising, by SAVER-X[16], produces more accurate estimates of the RNA transcript relative abundances for each cell. Compared to the raw counts, the denoised relative expression values have significantly improved correlation with their corresponding protein measurement (Figs. 1b, S3a, S4a, b). Yet, for some surface proteins, such as CD45RA, this correlation for denoised expression is still extremely low.

The production of a surface protein from its corresponding RNA transcript is a complicated process involving post-transcriptional modifications and transport[11], translation[12], post-translational modifications[13], and protein trafficking[14]. These processes depend on the state of the cell and the activities of other genes[9,15]. To learn the mapping from a cell's transcriptome to the relative abundance of a given set of surface proteins, cTP-net employs a multiple branch deep neural network (MB-DNN, Supplementary Fig. 1). Deep neural networks have recently shown success in modeling complex biological systems[17,18], and more importantly, allow good generalization across data sets[16,19]. Generalization performance is an important aspect of cTP-net, as we would like to perform imputation on tissues

that do not exactly match the training data in cell type composition. Details of the cTP-net model and training procedure, as well as of alternative models and procedures that we have tried, are in "Methods" section and Supplementary Note.

**Imputation accuracy evaluation via random holdout.** To examine imputation accuracy, we first consider the ideal case where imputation is conducted on cells of types that exactly match those in training data. For benchmarking, we used peripheral blood mononuclear cells (PBMCs) and cord blood mononuclear cells (CBMCs) processed by CITE-seq and REAP-seq[2,3], described in Supplementary Table 1. We employed holdout method, where the cells in each data set were randomly partitioned into two sets: a training set with 90% of the cells and a holdout set with the remaining 10% of the cells for validation (see the "Methods" section, Supplementary Fig. 2a). Each cell type is well represented in both the training and validation sets. Figs. 1b and S3a show that, for all proteins examined in the CITE-seq PBMC data, cTP-net imputed abundances have much higher correlation to the measured protein levels, as compared with the denoised and raw RNA counts of the corresponding genes. We obtained similar results for the CITE-seq CBMC and REAP-seq PBMC data sets (Supplementary Fig. 4a, b).

**Generalization accuracy to unseen cell types.** Next, we considered the generalization accuracy of cTP-net, testing whether it produces accurate imputations for cell types that are not present in the training set. For each of the high-level cell types in each data set in Supplementary Table 2, all cells of the given type are held out during training, and cTP-net, trained on the rest of the cells, was then used to impute protein abundances for the held out cells (see the "Methods" section, Supplementary Fig. 2b). We did this for each cell type and generated an "out-of-cell-type" prediction for every cell.

Across all benchmarking data sets and all cell types, these out-of-cell-type predictions still improve significantly upon the corresponding RNA counts while slightly inferior in accuracy to the traditional holdout validation predictions above (Figs. 2a and S4a). This indicates that cTP-net provides informative predictions on cell types not present during training, vastly improving upon using the corresponding mRNA transcript abundance as proxy for the protein level.

**Generalization accuracy across tissue and lab protocol.** To further examine the case where cell types in the training and test data are not perfectly aligned, we considered a scenario where the model is applied to perform imputation on a tissue that differs from the training data. We trained cTP-net on PBMCs and then applied it to perform imputation on CBMCs, and vice versa, using the data from Stoeckius et al.[3] (see the "Methods" section). Cord blood is expected to be enriched for stem cells and cells undergoing differentiation, whereas peripheral blood contains well-differentiated cell types, and thus the two populations are composed of different but related cell types. Figures 2a and S3b shows the result on training on CBMC and then imputing on PBMC. Imputing across tissue markedly improves the correlation to the measured protein level, as compared to the denoised RNA of the corresponding gene, but is worse than imputation produced by model trained on the same population. For practical use, we have trained a network using all cell populations combined, which indeed achieves better accuracy than a network trained on each separately (see the "Methods" section, Supplementary Figs. 3b and 4a, c). The weights for this network are publicly available at https://github.com/zhouzilu/cTPnet.

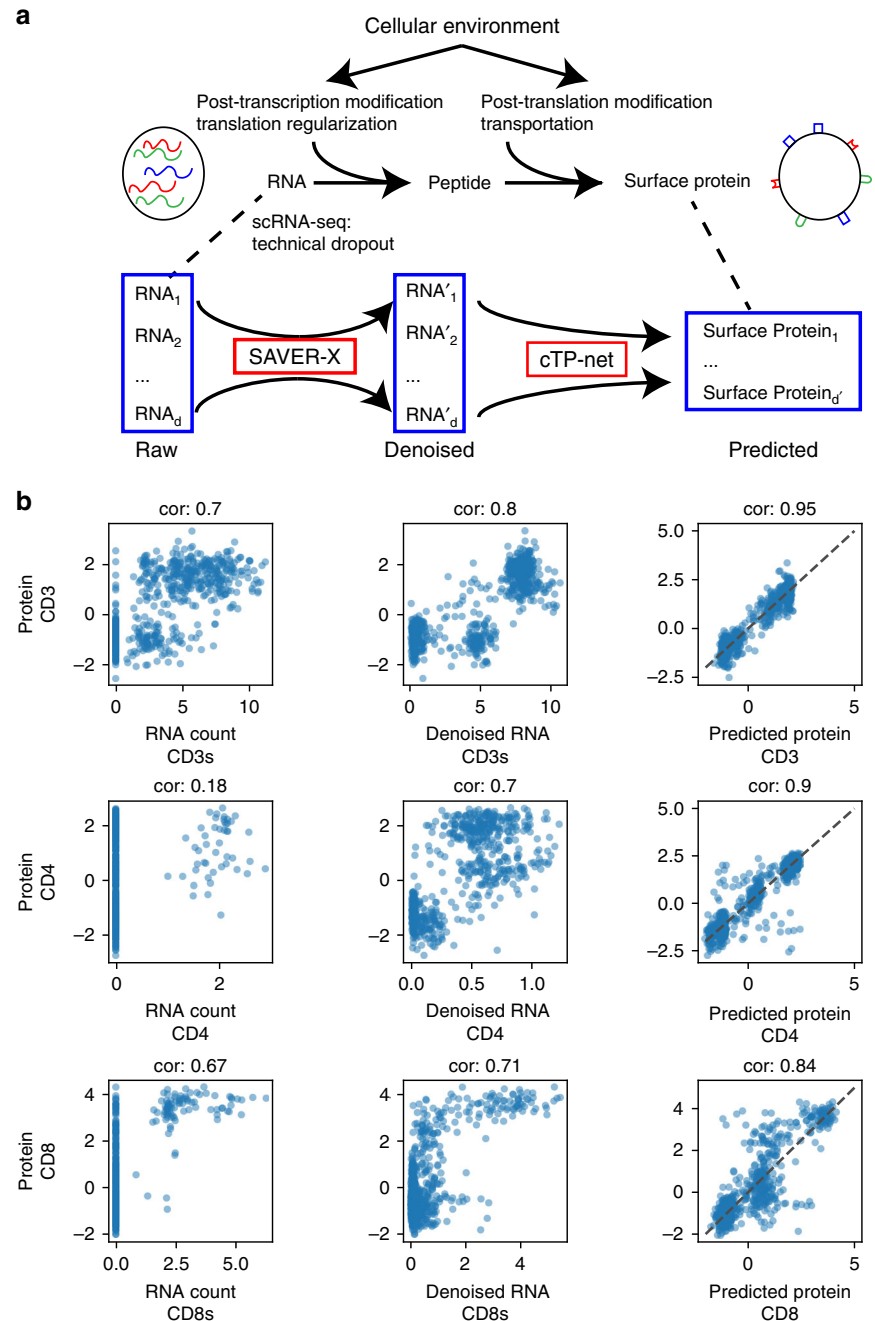

**Fig. 1 cTP-net analysis pipeline and imputation of example proteins. a** Overview of cTP-net analysis pipeline, which learns a mapping from the denoised scRNA-seq data to the relative abundance of surface proteins, capturing multi-gene features that reflect the cellular environment and related processes. **b** For three example proteins (CD3, CD4, and CD8), cross-cell scatter and correlation (cor) of CITE-seq measured abundances vs. (**1**) raw RNA count ("CD3s" and "CD8s" are sum of all genes that compose proteins CD3 and CD8, see Supplementary Table 6), (**2**) SAVER-X denoised RNA level, and (**3**) cTP-net predicted protein abundance.

We then tested whether cTP-net's predictions are sensitive to the laboratory protocol, and in particular, whether networks trained using CITE-seq data yields good predictions by REAP-seq's standard, and vice versa. Using a benchmarking design similar to above, we found that, in general, cTP-net maintains good generalization power across these two protocols (Figs. 2a and S3b).

**Imputation accuracy comparison to Seurat v3.** Seurat v3 anchor transfer[20] is a recent approach that uses cell alignment between data sets to impute features for single cell data. For comparison, we applied Seurat v3 anchor transfer to the holdout validation and out-of-cell-type benchmarking scenarios above (see the "Methods" section). In the validation scenario, we found the performance of cTP-net and Seurat v3 to be comparable, with cTP-net slightly better, as both methods can estimate protein abundance by utilizing marker genes to identify the cell types. cTP-net, however, vastly improves upon Seurat in the out-of-cell-type scenario (Figs. 2a and S5a). This is because cTP-net's neural network, trained across a diversity of cell types, learns a direct transcriptome-protein mapping that can more flexibly generalize

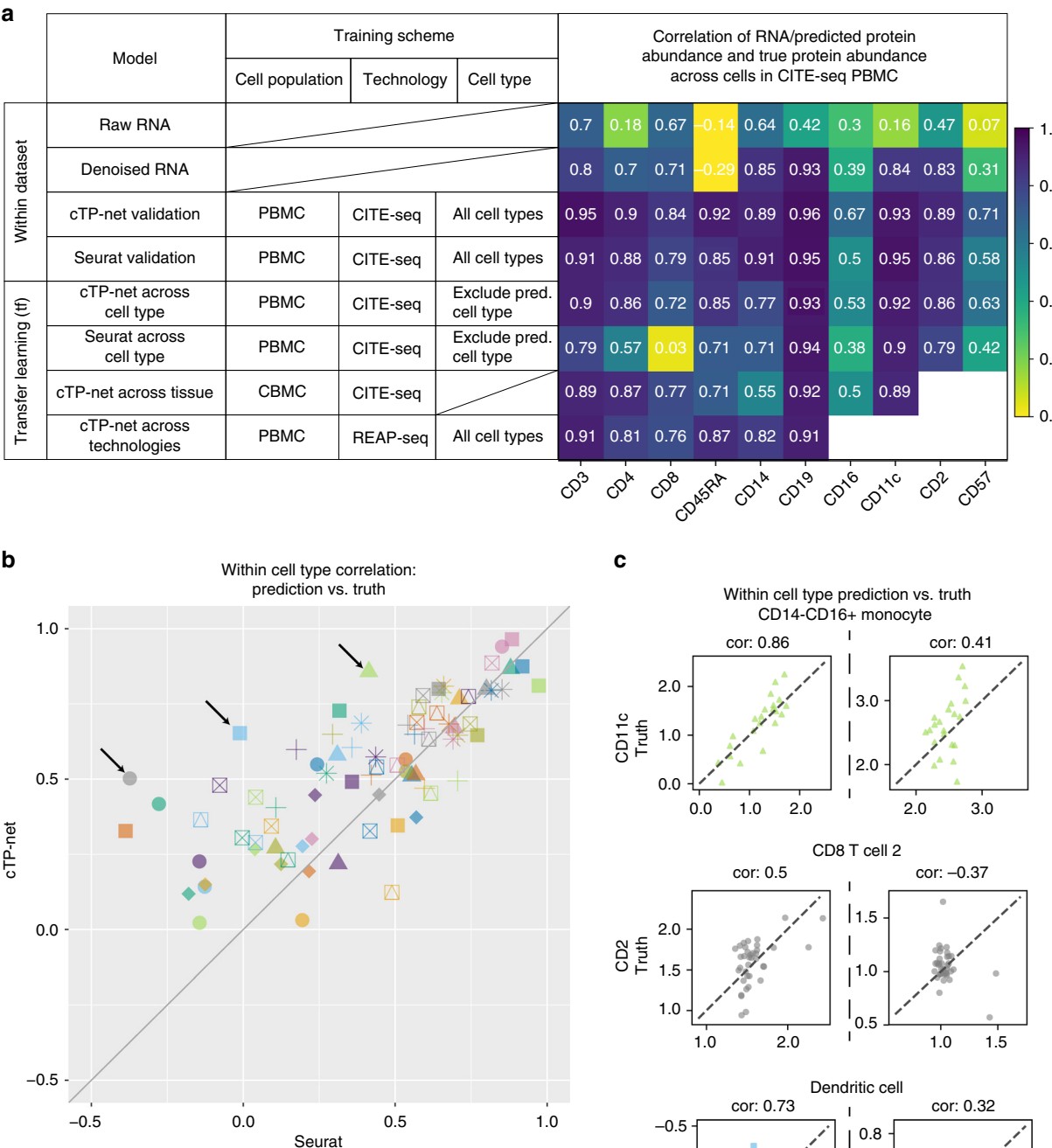

**Fig. 2 Benchmark evaluation on CITE-seq PBMC data. a** Benchmark evaluation of cTP-net on CITE-seq PBMC data, with comparisons to Seurat v3, in validation, across cell type, across tissue, and across technology scenarios. The table on the left shows the training scheme of each test, the heatmap shows correlations with actual measured protein abundances. **b** Within cell type correlations between imputed and measured protein abundance on the CITE-seq PBMC data, Seurat v3 versus cTP-net. Each point (color and shape pair) indicates a cell type and surface protein pair, where the x-axis is correlation between actual measured abundance and Seurat imputation and y-axis is the correlation between actual measured abundance and cTP-net imputation. **c** Scatter of imputed versus measured abundance for the three (surface protein, cell type) pairs marked by arrows in **b**: CD11c in CD14−CD16+ monocytes, CD2 in CD8 T cells, and CD19 in dendritic cells.

to unseen cell types, while Seurat v3 depends on a nearest-neighbor method that can only sample from the training dataset. As shown by the cross-population and out-of-cell-type benchmarking above, cTP-net does not require direct congruence of cell types across training and test sets.

In addition to predictions on unseen cell type, cTP-net also improves upon the existing state-of-the-art in capturing within cell-type variation in protein abundance. As expected, within cell-type variation is harder to predict, but cTP-net's imputations nevertheless achieve high correlations with measured protein abundance for a subset of proteins and cell types (Supplementary Figs. 3c and 4d). Compared to Seurat v3, cTP-net's imputations align more accurately with measured protein levels when zoomed into cells of the same type (Figs. 2b and S5b); see Fig. 2c, for example, CD11c in CD14−CD16+ monocytes, CD2 in CD8 T cells, and CD16 in dendritic cells. All of these surface proteins have important biological function in the corresponding cell types, as CD11c helps trigger respiratory burst in monocyte[21], CD2 co-stimulates molecule on T cells[22] and CD16 differentiate DC subpopulation[23]. The learning of such within-type heterogeneity gives cTP-net the potential to attain higher resolution in the discovery and labeling of cell states.

**Network interpretation and feature importance**. What types of features are being used by cTP-net to form its imputation? To interpret the network, we conducted a permutation-based interpolation analysis, which calculates a permutation feature importance for each protein–gene pair (see the "Methods" section, Supplementary Fig. 6a). Interpolation can be done using all cells, or cells of a specific type, the latter allowing us to probe relationships that may be specific to a given cell type. Applying this analysis to cTP-net trained on PBMC, we found that, at the level of the general population that includes all cell types, the most important genes for the prediction of each protein are those that exhibit the highest cell-type specificity in expression (Supplementary Table 3). This is because most of these surface proteins are cell type markers, and thus when cells of all types are pooled together, "cell type" is the key latent variable that underlies their heterogeneity. In addition, as cell-type markers are usually redundant and predictable by other genes, the model still performs well after removing corresponding surface protein genes during training (Supplementary Tables 5 and 6). Within cell type interpolation, on the other hand, reveals genes related to RNA processing, RNA binding, protein localization, and biosynthetic processes, in addition to immune-related genes that differentiate the immune cell sub-types (Supplementary Table 4). This analysis shows that cTP-net combines different types of features, both cell type markers and genes involved in RNA to protein conversion and transport, to achieve multiscale imputation accuracy.

In addition, we analyzed the bottleneck layer with 128 nodes before the network branched out to the protein-specific layers. We performed dimension reduction (UMAP) directly on the bottleneck layer intermediate output of 7000 PBMCs from CITE-seq. Supplementary Fig. 6b shows that the cells are cleanly separated into different clusters, representing cell types as well as gradients in surface protein abundance. This confirms that the bottleneck layer captures the essential information on cell stages and transitions, and that each subsequent individual branch then predicts its corresponding protein's abundance.

**Application to Human Cell Atlas**. Having benchmarked cTP-net's generalization accuracy across immune cell types, tissues, and technologies, we then applied the network trained on the combined CITE-seq dataset of PBMCs,CBMCs, and bone marrow mononuclear cells (BMMCs)[3,20] to perform imputation for

the Human Cell Atlas CBMC and BMMC data sets (Supplementary Table 1). Figure 3 shows the raw RNA count and predicted surface protein abundance for 24 markers across 6023 BMMCs from sample MantonBM1 and 4176 CBMCs from sample MantonCB1. (Similar plots for the other 7 BMMC and 7 CBMC samples are shown in Supplementary Figs. 8 and 9). Similar to what was observed for actual measured protein abundances in the CITE-seq and REAP-seq studies, the imputed protein levels differ markedly from the RNA expression of its corresponding gene, displaying higher contrast across cell types and higher uniformity within cell type. Thus, the imputed protein levels serve as interpretable intermediate features for the identification and labeling of cell states, defining cell subtypes more clearly than the RNA levels of the corresponding marker genes. For example, imputed CD4 and CD8 levels separate CD4+ T cells from CD8+ T cells with high confidence. Further separation of naïve T cells to memory T cells can be achieved through imputed CD45RA/CD45RO abundance, as CD45RA is a naïve antigen and CD45RO is a memory antigen. Consistent with flow cytometry data, the large majority of CB T cells are naïve, whereas the BM T cell population is more diverse[24]. Also, for BM B cells that have high imputed CD19 levels, cTP-net allows us to confidently distinguish the Pre.B (CD38+, CD127+), immature B (CD38+, CD79b+), memory B (CD27+), and naïve B cells (CD27−), whose immunophenotypes have been well characterized[25].

In addition, consider natural killer (NK) cells, in which the proteins CD56 and CD16 serve as indicators for immunostimulatory effector functions, including an efficient cytotoxic capacity[26,27]. We observe an opposing gradient of imputed CD56 and CD16 levels within transcriptomically derived NK cell clusters that reveal CD56$^{bright}$ and CD56$^{dim}$ subsets, coherent with previous studies[3] (Fig. 2f, Supplementary Fig. 10, F-test: p-value = 1.667e−15). This pattern is not found in RNA abundances due to low expression (F-test: p-value = 0.9377). Between CD56$^{bright}$ and CD56$^{dim}$ subsets, 7 out of 10 of previously studied differentially expressed genes are significant in the single cell analysis (Fisher test: p-value = 1.07e−04)[3,28,29]. This gradient in CD56 and CD16, where decrease in CD56 is accompanied by increase in CD16, is replicated across the 8 CBMC and 8 BMMC samples in HCA (Supplementary Figs. 8–10).

Consider also the case of CD57, which is a marker for terminally differentiated "senescent" cells in the T and NK cell types. The imputed level of CD57 is lower in CBMCs (fetus's blood), and rises in BMMCs (95% quantile: bootstrap p-value < 1e−6). This is consistent with expectation since CD57+ NK cell and T cell populations grow after birth and with ageing[30–32] (Supplementary Figs. 8 and 9).

These results demonstrate how cTP-net, trained on a combination of PBMCs, CBMCs, and BMMCs, can impute cell type, cell stage, and tissue-specific protein signatures in new data without explicitly being given the tissue of origin.

**Application to AML**. We further apply cTP-net to an AML data set from van Galen et al.[33]. AML is a heterogeneous disease where the diversity of malignant cell types partially recapitulates the stages of myeloid development. Mapping the malignant cells in AML to the differentiation stage of their cell of origin strongly impacts tumor prognosis and treatment, as malignant cells that originate from earlier stage progenitors have higher risk of relapse[34,35]. In the original paper, the authors sequenced 7698 cells from five healthy donors to build a reference map of cell types during myeloid development, and then mapped 30,712 cells from 16 AML patients across multiple time points to this reference to identify the differentiation stage of the malignant cells. Here, by imputing 24 immunophenotype markers with cTP-net,

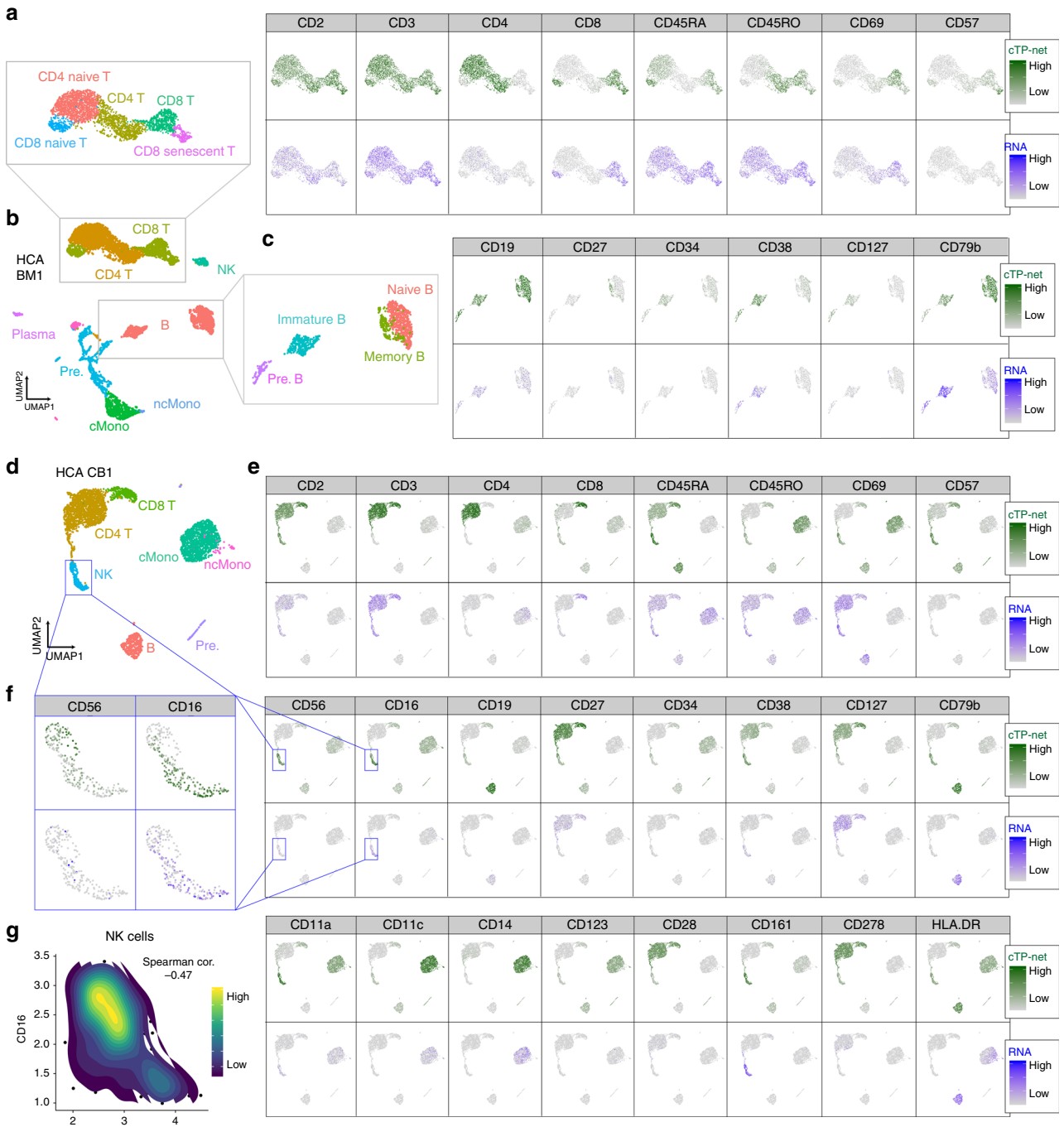

**Fig. 3 Imputation results analysis on Human Cell Atlas data sets. a** Left panel: UMAP visualization of MantonBM1 BMMCs T cell subpopulation based on RNA expression, colored by cell type. CD4 T: mature CD4+ T cells; mature CD8 T: CD8+ T cells; naïve CD4 T: naïve CD4+ T cells; naïve CD8 T: naïve CD8+ T cells; CD8 senescent T: CD8+ senescent T cells. Right: Related imputed protein abundance and RNA expression of its corresponding gene. **b** UMAP visualization of MantonBM1 BMMCs based on RNA expression, colored by cell type. B: B cells; CD4 T: CD4+ T cells; CD8 T: CD8+ T cells; cMono: classical monocyte; ncMono: non-classical monocyte; NK: natural killer cells; Pre.: precursors; Plasma: plasma cells. **c** Left panel: UMAP visualization of MantonBM1 BMMCs B cell subpopulation based on RNA expression, colored by cell type. Pre.B: B cell precursors; immature B: immature B cells; memory B: memory B cells; naïve B: naïve B cells. Right panel: Related imputed protein abundance and RNA expression of its corresponding gene. **d** UMAP visualization of MantonCB1 CBMCs based on RNA expression, colored by cell type. **e** cTP-net imputed protein abundance and RNA read count of its corresponding gene for 24 surface proteins. **f** UMAP visualization of MantonCB1 CBMCs NK cell subpopulation colored by CD56 and CD16 imputed protein abundance and RNA read count. Reverse gradient is observed in cTP-net prediction but not in the read count for its corresponding RNA. **g** Contour plot of cells based on imputed CD56 and CD16 abundance in NK cell populations. Strong negative correlation (Spearman correlation = −0.47) with two subpopulation observed.

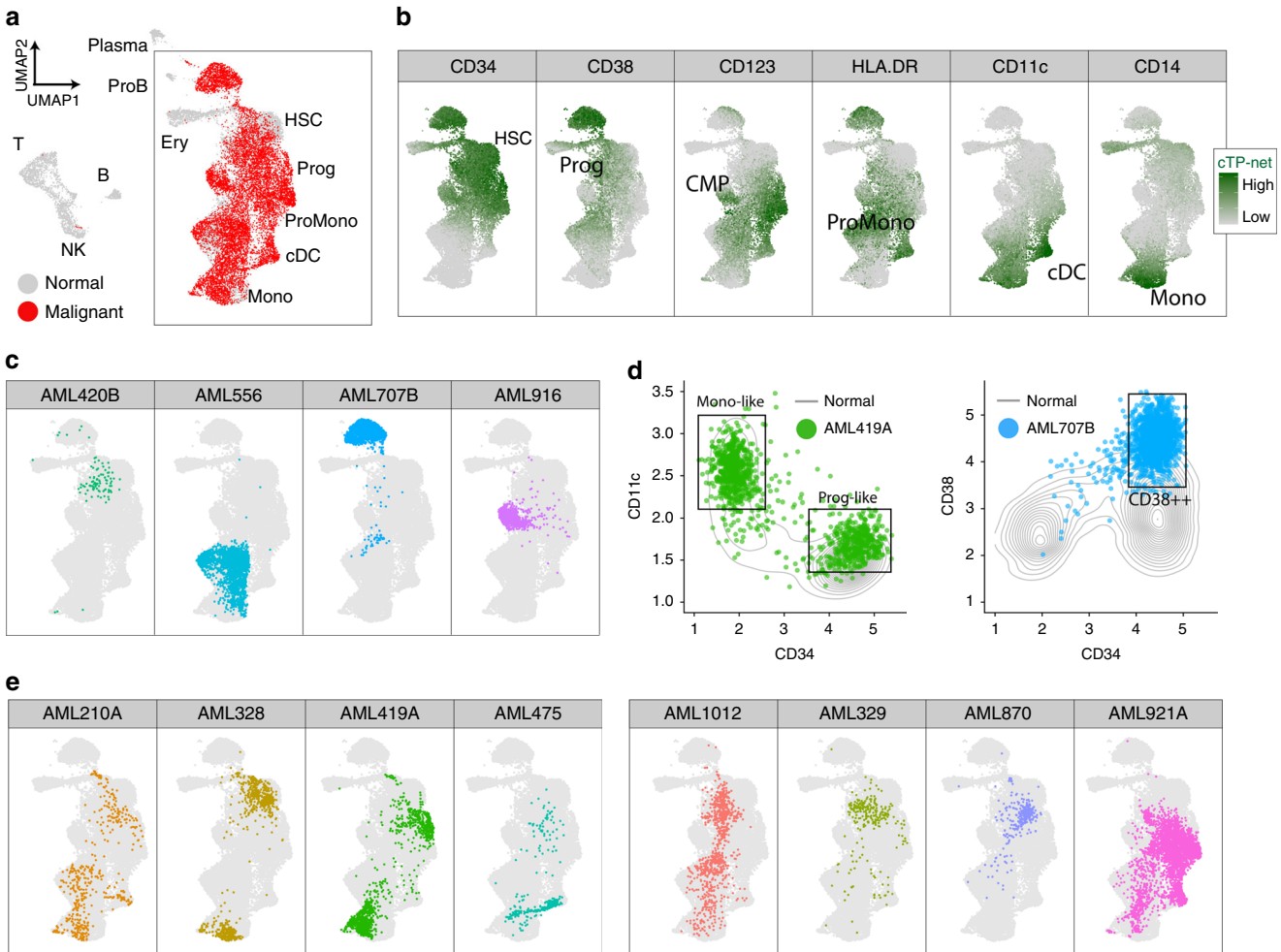

**Fig. 4 Imputation results analysis on acute myeloid leukemia data sets. a** UMAP visualization of normal cells and malignant cells from 12 AML samples at Day 0 based on imputed protein abundance (red: malignant cells; gray: normal cells). **b** UMAP visualization of the myeloid trajectory. cTP-net imputed protein abundance of markers that perfectly recapitulate the myeloid development. **c**, **e**, **f** UMAP visualization of the myeloid trajectory with corresponding malignant cells from AML sample highlighted. **d** Plot of normal cells (gray contour) and AML malignant cells (dots) based on imputed protein expression.

we can directly characterize the differentiation stage of cell-of-origin for the malignant cells.

Figure 4a is a UMAP plot based on imputed surface protein abundance of five normal BMs and 12 Day 0 samples from AML patients. The majority of the malignant cells as identified in the original paper reside on the right half of the plot, which recapitulate the myeloid differentiation trajectory as revealed by the imputed values of canonical protein markers (Fig. 4b): From CD34+ progenitors to CD38+ CD123+ cells in transition to CD11c+ and CD14+ mature monocytes[36]. All of the malignant cells have imputed protein values that place them along this monocyte lineage. Using the transcriptome for visualization, on the other hand, reveals large batch effects across samples, due to both technical batch and biological differences (Supplementary Fig. 11). Thus, unlike the imputed protein data, the transcriptomic data cannot be directly combined without alignment.

Based on the trajectory revealed by the imputed protein levels, we can determine the differentiation cell stage(s) for the malignant cells of each tumor, according to which the 12 AML patients can be divided into three categories: (1) AMLs of single differentiation stage (AML420B, AML556, AML707B, and AML916; Fig. 4c), (2) AMLs of two differentiation stages (AML210A, AML328, AML419A, and AML475; Fig. 4e), and (3) AMLs of many differentiation stages (AML1012, AML329, AML870, and AML921A; Fig. 4f). This stage assignment is consistent with the original study[33]. For example, AML419A harbors two malignant cell types at opposite ends of the monocyte differentiation axis, distinguished by imputed CD34 and CD11c levels as CD34+ CD11c− indicates progenitor-like and CD34−CD11c+ indicates differentiated monocyte-like cells (Fig. 4d, e). AML707B, which carries a RUNX1/RUNX1T1 fusion, consists of cells of a specific cell stage that is distinct from the normal myeloid trajectory (Fig. 4c). Such unique cell cluster was due to hyper CD38 level in surface protein prediction (Fig. 4d). Such hyper-CD38 levels have been reported in AMLs with RUNX1/RUNX1T1 fusion[37–39] and recent studies have also shown that CD38 can be a potential target for adult AML[40,41].

In this example, the imputed protein levels served as useful features for trajectory visualization. This analysis also indicates that even though cTP-net is currently trained only on normal immune cells, it can reveal disease-specific signatures in malignant cells and the imputed protein levels are useful for characterizing tumor phenotypes.

## Discussion

Taken together, our results demonstrate that cTP-net can leverage existing CITE-seq and REAP-seq datasets to predict surface protein relative abundances for new scRNA-seq data sets, and that the predictions generalize to cell types that are absent from,

but related to those in the training data. cTP-net was benchmarked on PBMC and CBMC immune cells, showing good generalization across tissues and technical protocols. On Human Cell Atlas data, we show that the imputed surface protein levels allow easy assignment of cells to known cell types, as well as the revealing of intra-cell type gradients. We then demonstrate that, even though cTP-net used only immune cells from healthy individuals for training, it is able to impute immunophenotypes for malignant cells from AML, and that these immunophenotypes allow placement of the cells along the myeloid differentiation trajectory. Furthermore, we show that cTP-net is able to impute protein signatures in the malignant cells that are disease specific and that are not easily detectable from the transcriptomic counts.

SAVER-X serves an important role in the training procedure of cTP-net. As shown in Supplementary Table 5, without SAVER-X denoising, the cTP-net prediction performance retracts by 0.02 in correlation, more significant than any other parameter tweaks. This discrepancy in performance is due to: (1) SAVER-X makes use of the noise model to obtain estimates of the true RNA counts. This helps cTP-net learn the underlining relationship between true RNA counts and protein level, rather than the noisy raw counts and protein levels, which varies more across data sets and thus does not generalize well. (2) By denoising the scRNA-seq, the input for learning the RNA–protein relationship is less sparse. Manifold learning on a more continuous input space usually works better[42,43]. (3) Comparing to other autoencoder-based denoising method, SAVER-X performs Bayesian shrinkage on top of autoencoder framework to prevent over-imputation (over-smoothing)[16,44].

Despite these promising results, cTP-net has limitations: (1) cTP-net can only apply to count-based expression input (UMI-based). CITE-seq data with TPM and RPKM expression metric is not available for testing. Thus, the prediction accuracy is unknown. (2) The generalization ability of cTP-net to unrelated cell types has limitations. Even though the final cTP-net model, trained on immune cells, has good results on immune cells from diverse settings, we have not tried to perform imputation of these immune-related markers on cells that are not of the hematopoietic lineage.

With the accumulation of publicly available CITE-seq and REAP-seq data across diverse proteins, cell types, and conditions, cTP-net can be retrained to accommodate more protein targets and improve in generalization accuracy. The possibility of such cross-omic transfer learning underscores the need for more diverse multi-omic cell atlases, and demonstrate how such resources can be used to enhance future studies. The cTP-net package is available both in Python and R at https://github.com/zhouzilu/cTPnet.

## Methods

**Data sets and pre-processing.** Supplementary Table 1 summarizes the five data sets analyzed in this study: CITE-PBMC, CITE-CBMC, REAP-PBMC, HCA-CBMC, and HCA-BMMC. Among these, CITE-PBMC, CITE-CBMC, and REAP-PBMC have paired scRNA-seq and surface protein counts, while HCA-CBMC and HCA-BMMC have only scRNA-seq counts. For all scRNA-seq data sets, low-quality gene (<10 counts across cells) and low-quality cells (<200 genes detected) are removed, and the count matrix ($C$) for all remaining cells and genes is used as input for denoising. scRNA data denoising was performed with SAVER-X using default parameters. Denoised counts ($\Lambda$) were further transformed with Seurat default LogNormalize function

$$X_{ij} = \log\left(\frac{\Lambda_{ij} * 10,000}{m_j}\right) \quad (1)$$

where $\Lambda_{ij}$ is the denoised molecule count of gene $i$ in cell $j$, and $m_j$ is the sum of all molecule counts of cell $j$. The normalized denoised count matrix $X$ is the training input for the subsequent multiple branch neural network. For the surface protein counts, we adopted the relative abundance transformation from Stoeckius et al.[3].

For each cell $c$

$$y_c = \left[\ln\left(\frac{p_{1c}}{g(\mathbf{p}_c)}\right), \ln\left(\frac{p_{2c}}{g(\mathbf{p}_c)}\right) \ldots \ln\left(\frac{p_{dc}}{g(\mathbf{p}_c)}\right)\right] \quad (2)$$

where $\mathbf{p}_c$ is vector of antibody-derived tags (ADT) counts, and $g(\mathbf{P}_c)$ is the geometric mean of $\mathbf{p}_c$. The network trained using this transformed relative protein abundance as the response vector yields better prediction accuracy than the network trained using raw protein barcode counts.

**cTP-net neural network structure and training parameters.** Supplementary Fig. 1 shows the structure of cTP-net. Here, we have a normalized expression matrix $\mathbf{X}$ of $N$ cells and $D$ genes, and a normalized protein abundance matrix $\mathbf{Y}$ of the same $N$ cells and $d$ surface proteins. Let us denote cTP-net as a function $F$ that maps from $\mathbb{R}^D$ to $\mathbb{R}^d$. Starting from the input layer, with dimension equals to number of genes $D$, the first internal layer has dimension 1000, followed by a second internal layer with dimension 128. These two layers are designed to learn and encode features that are shared across proteins, such as features that are informative for cell type, cell state, and common processes such as cell cycle. The remaining layers are protein specific, with 64 nodes for each protein that feed into a one node output layer giving the imputed value. All layers except the last layer are fully connected (FC) with rectified linear unit (ReLU) activation function[45], while the last layer is a FC layer with identity activation function for output. The objective function here is

$$\underset{F}{\mathrm{argmin}}|\mathbf{Y} - F(\mathbf{X})|_1 \quad (3)$$

where the loss is L1 norm. The objective function was optimized stochastically with Adam[46] with learning rate set to 10e−5 for 139 epochs (cross-validation). Other variations of cTP-net, which we found to have inferior performance, are illustrated in more details in Supplementary Note.

**Benchmarking procedure.** Supplementary Fig. 2a shows the validation set testing procedure. Given limited amount of data, we keep only 10% of the cells as the testing set, and use the other 90% of the cells for training. The optimal model was selected based on the testing error.

We perform the out-of-cell type prediction based on Supplementary Fig. 2b. This procedure mimics cross-validation, except that, instead of selecting the test set cells randomly, we partition the cells by their cell types. Iteratively, we designate all cells of a given cell type for testing and use the remaining cells for training. We then perform prediction on the hold-out cell type using the model trained on all other cell types. In the end, every cell has been tested once and has the corresponding predictions. In the benchmark against the validation set testing procedure, we limit comparisons to the same cells that were in the validation set in the holdout scheme to account for variations between subsets.

To apply the models we trained in validation set testing procedure to different cell populations and technologies, the inputs have to be in the same feature space. Even though all data sets considered are from human cells, the list of genes differs between experiments and technologies. Genes that are in the training data but not in the testing data are filled with zeros. Because cTP-net utilizes overrepresented number of genes to predict the surface proteins level, having a small number of genes missing has little effect on the performance. After prediction, we selected only the shared proteins between two data sets for comparison.

**cTP-net interpolation.** To better interpret the relationships that the neural network is learning, we developed a permutation-based interpolation scheme that can calculate an influence score $epi$ for each gene in the imputation of each protein (Supplementary Fig. 6). The idea is to assess how much changing the expression value of certain genes in the training data affects the training errors for a given model $F$. In each epoch, we interpolate all of the genes in a stochastic manner. Let us denote $\mathbf{X}$ as the expression matrix ($N$ by $G$ matrix, where $N$ is the number of cells and $G$ is number of genes), $\mathbf{Y}$ as protein abundance matrix and $L$ as the loss function. The algorithm goes as follows (Supplementary Fig. 6):

Estimate the original model error $\epsilon^{\mathrm{orig}} = L(\mathbf{Y}, F(\mathbf{X}))$.

Sampling batch of genes denote by. Generate expression matrix $\mathbf{X}^{\mathrm{perm}}$ by permuting genes in $gs$ in the data $\mathbf{X}$. This breaks the association between $gs$ and protein abundance $\mathbf{Y}$, i.e. the cell order within $gs$ does not coordinate with protein abundance $\mathbf{Y}$.

Estimate error $\epsilon^{\mathrm{perm}} = L(\mathbf{Y}, F(\mathbf{X}^{\mathrm{perm}}))$ based on the predictions of the permuted data.

Calculate permutation feature importance $\Delta_{\mathrm{gs}} = |\epsilon^{\mathrm{orig}} - \epsilon^{\mathrm{perm}}|$ of gene set $gs$ to this model $F$.

We set batch size as 100 with 500 epochs. Furthermore, by picking different cells to interpolate, we could identify gene influence score in different cell types. For example, if matrix $\mathbf{X}$ belongs to a given cell type, the cell type-specific genes are consistent across cells of the given cell type, and thus, the permutation will not influence these genes. Genes that influence the surface protein abundance within the cell type, such as cell cycle genes and protein synthesis genes, tend to be rewarded with high influence scores in such a cell-type-specific interpolation analysis.

For the top 100 highest influence scored genes from the following scenarios in CITE-PBMC: (1) CD45RA in CD14−CD16+ monocytes, (2) CD11c in CD14−CD16+ monocytes, (3) CD45RA in CD8 T cells, (4) CD45RA in CD4 T cells, (5) CD11c in CD14+CD16+ monocytes, (6) CD45RA in dendritic cells, and (7) CD11c in dendritic cells, we employed a Gene Ontology analysis[47] which identify top 10 pathways based on GO gene sets with FDR $q$-value < 0.05 as significant (Supplementary Table 4).

**Seurat anchor-transfer analysis**. We compared cTP-net with an anchor-based transfer-learning method developed in Seurat v3[20]. For Seurat v3, RNA count data are normalized by LogNormalization method, while surface protein counts are normalized by centered log-ratio (CLR) method. In validation test setting, we used the same cells for training and testing as in cTP-net so as to be directly comparable to cTP-net. For out-of-cell type prediction, default parameters did not work for several cell types in anchor-transfer step, because, for those cell types, there are few anchors shared between the training and testing sets. To overcome this, we reduced the number of anchors iteratively until the function ran successfully.

**HCA data analysis**. HCA RNA-seq data sets are pre-processed as discussed above, resulting in log-normalized denoised values. We applied default pipeline of Seurat and generated UMAP plot for both data sets (Supplementary Fig. 7). Cells are clearly clustered by individuals, indicating strong batch effects. As a result, the following analysis was performed on cells of each individual. Major cell types were determined by known markers.

From the log-normalized denoised expression value, we predict the surface protein abundance with cTP-net model trained jointly on CITE-seq PBMC, CBMC, and BMMC data sets. We embedded 24 surface protein abundance across 16 individuals on t-SNE plot, showing consistent results with cell type information (Supplementary Figs. 8 and 9).

**Reporting summary**. Further information on research design is available in the Nature Research Reporting Summary linked to this article.

## Data availability
Public datasets for training and evaluating cTP-net can be found at National Center for Biotechnology Information Gene Expression Omnibus (GEO) under accession number GSE100866, GSE100501, and GSE128639 respectively.

## Code availability
cTP-net package are publicly available as both an open-source R package at https://github.com/zhouzilu/cTPnet with license GPL-3.0 and an open-source python package at https://github.com/zhouzilu/ctpnetpy with license GPL-3.0.

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

## Acknowledgements

We would like to thank the National Institutes of Health for the award 5R01-HG006137 (for Z.Z. and N.R.Z.), award 1U2CCA233285-01 (for N.R.Z.), the National Science Foundation for the award DMS-1562665 (to J.W., N.R.Z.), and the Wharton Dean's Fund for Post-doctoral Research (to J.W.).

## Author contributions

Z.Z. and N.R.Z. conceptualized the study and planned the case studies. Z.Z. designed the model, developed the algorithm, implemented the cTP-net software, and led the data analysis. C.Y. helped in CITE-seq and REAP-seq data denoising and cell type labeling. J.W. helped with model design and Human Cell Atlas data analysis. Z.Z. and N.R.Z. wrote the paper with feedback from C.Y. and J.W.

## Competing interests

The authors declare no competing interests.
