## [Peer Review File · Nature Communications]

Reviewers' comments:

Reviewer #1 (Remarks to the Author):

Zhou et al describe a neural network approach to impute protein expression from scRNAseq data called (cTP-net). The authors demonstrate imputation accuracy in three of publicly available data sets. I find the manuscript well written and easy to follow. However, in the current state it may be better suited for a more focused journal. Most importantly the manuscript's demonstration of how one would use cTP-net in the analysis is insufficient. A more detailed list of concerns is outlined below:

Major concerns:

1. The authors describe an exemplary application of cTP-net using the HCA CBMC and BMMC data. However, the analysis highlighting CD56+/- NK subpopulations seems very "cherry picked" and is not convincing. The authors describe the visual expression pattern of a single protein in a tSNE plot. A systematic analysis is required to demonstrate that cTP-net adds meaningful value to biological analysis/interpretation, ideally over multiple data sets/scenarios. I.e. do these two subpopulations form separate clusters when including the imputed protein expression as a feature? Is the "opposing gradient" statistically significant? Across all donors? Does differential expression analysis on the imputed protein values reveal the right markers?
2. From a deep learning perspective it is not clear why the denoising step is required. At first I thought it was required to reduce the dimensionality of the input space but (as far as I understood) the denoised matrix has the same dimensions as the original gene expression matrix. It find it hard to believe that training the network directly on the original gene expression input will not achieve equal performance (or better). Table S5 contains the performance summary of a number of different models. However, direct comparison with and without the SAVER-X denoising is not provided.
3. The authors should consider training cTP-net on the latent space of SAVER-X. Could this improve the model by decreasing dimensionality?
4. Throughout the manuscript accuracies and correlation coefficients for different models/approaches are missing standard deviations/errors. As the differences are small for some comparisons it is important for the reader to get an understanding of the uncertainties derived from running the model multiple times.
5. How does cTP-net deal with over-imputation? What happens if my input is a completely unrelated celltype? What protein predictions will I get? Can the authors provide any guidance on this potential issue for users?
6. As the authors correctly point out in their SAVER-X manuscript different scRNAseq technologies differ in the underlying model assumptions (NB vs ZINB). Can cTP-net successfully predict across droplet and well based scRNAseq technologies?
7. I was intrigued by the interpolation analysis but would like to see more analyses towards understanding the properties of the network. I.e. if data was clustered using the activations in the 128 neuron layer, would this return the various cell types/protein expression clusters?

Minor concerns:

1. Additional information with respect to training is missing. I.e. number of epochs, early stopping etc?
2. p. 3 typo - "... these out-of-cell-type predictions are, as expected, are still greatly improve upon the corresponding RNAs while slightly inferior in accuracy to the traditional holdout validation ..."

Reviewer #2 (Remarks to the Author):

In this paper the authors introduce cTP-net (single cell Transcriptome to Protein prediction with deep neural network), a deep learning tool which leverages single-cell sequencing data to predict cell-surface protein data. The tool shows promising results, especially with regards to its accuracy

when translating to novel cell types. Furthermore, as multi-omic training data continue to become more prevalent, the model will continue to improve in its precision and hence applicability. This being said, I feel that the paper needs to be significantly improved before it can be published. In particular, the methodology should be better motivated from a biological point of view. The model should be evaluated against cell-type prediction model since proteins are often used to predict cell-types. The denoising part needs to be better motivated.

Major comments.

- The organization of the paper is slightly scattered. More sections and headers would help. Specifically the paper lacks an introduction section and a discussion section.

- The paper lacks discussion of how the model predictions can be used in a broader context. Perhaps presenting an example analysis workflow where this method is integral would make its usefulness more clear. It's not clear what one would do with the predicted protein expression. It would be good to present a motivating example. More specifically, it seems that the main application here is in predicting cell type. In this case, one wonders whether it is really necessary to predict cell types.

- It's not clear to me what one needs to perform denoising before prediction. One would expect the model to be able to learn this as part of the process. So more comparisons need to be made with and without imputation. And why SAVER-X? Does it matter, could you use another tool?

- 109-119. It would be nice to explicitly separate the two ways described. From what I understand, there are two steps a model takes when computing protein expression: (1) identify cell-type markers to identify cell type, which helps estimate protein counts, and/or (2) use GEX patterns to directly identify protein expression. I think the argument in the paper is that Seurat v3 fails to do the second step, whereas cTP-net does both. Then this benefit of cTP-net would give it an extra edge over Seurat particularly when it comes to variations that are not explained through cell-type differentiation.

- The paper does not explicitly show the model's performance compared to purely using cell-type marker genes. It is not clear just how much of the performance can be easily explained by good use of cell-type marker genes. It would be good to use the approach with and without marker genes. For example, when predicting CD3 and CD4, the respective genes should be excluded.

- 129-130. I don't understand how this demonstrates that cTP-net uses both types of gene in its predictions. From what I understand, you are again saying that cTP-net can use GEX patterns for more than just determining cell type, but also to directly predict protein expression. To me this point is just a restatement of the previous paragraph, so I think that the connection between the two points could be made more clear.

- 140. Related to the points above many tools have already been published to predict cell type from gene expression. So your proposed approach should be compared to these tools when the end goal is cell type.

- 281-292. Why use these particular settings? Can you provide an intuitive explanation? See comment for table S5. I think it would be good to perform a better sensitivity analysis to see how the parameters affect performance.

- 318-329. I find this section very confusing. Specifically, I cannot understand the line 'for genes within *gs*, cell labels were permuted'. However, this whole section could use more/clearer explanation.

Minor comments:

- Fig 2. The aesthetics are cluttered. It might be better to leave a part out or break this into separate figures.
- S1. It is not clear how the correlation is calculated.
- S8/S9. It might be nice to also show the actual counts for each protein, or at least briefly address the lack thereof. I am curious if there are cases where cTP-net generated protein values differ significantly from the actual counts.
- Table S5. Why were these particular changes chosen? I think few people would argue that the MB structure is bad, but perhaps other changes would make a difference. Here are other adaptations I might have liked to see tested, in order of importance:
 - dropout layer (probably before the first hidden layer)
 - activation functions (they only use relu and don't consider any others)
 - number of layers (this would help model nonlinear parts of the function)
 - learning rate
 - number of nodes (what about reducing the bottleneck?)
 - others? (loss function/optimization function/preprocessing)
- Figure 2A, why is the correlation for CD16 so weak?
- Figure 2C is unclear, what are we looking at? What does CD19, DC means? CD19 is not expressed in DCs as it is a B-cell surface marker.

Minor fixes:

- 30. become → becomes
- 61. CBMC's are cord blood mononuclear cells (not cordical)
- 77. grammar
- 80-81. it seems misleading to keep referencing this proxy as though people expect it to be highly accurate.
- 290. "where the loss [is] l1 norm"
- S8/S9 legend → nature killer cells should be natural killer cells

Minorer fixes:

- grammar on 69/70 is awkward
- 289. I am not familiar with the notation (the 1's on the right)
- S9 → NK label missing in BM2 figure

Reviewer #1 (Remarks to the Author):

Zhou et al describe a neural network approach to impute protein expression from scRNAseq data called (cTP-net). The authors demonstrate imputation accuracy in three of publicly available data sets. I find the manuscript well written and easy to follow. However, in the current state it may be better suited for a more focused journal. Most importantly the manuscript's demonstration of how one would use cTP-net in the analysis is insufficient. A more detailed list of concerns is outlined below:

Thank you for taking the time to review our paper, we appreciate all of your suggestions. On the broadness of interest in this work: The prediction of surface proteins links new scRNA-seq immune data sets with classical immunophenotypes, as surface proteins have played a central role in cell state classification and spatiotemporal imaging for immune cells. Immune cells play key roles in most tissues, and thus we believe this work should be useful for most researchers working with scRNA-seq data. Also, although cTP-net focuses on RNA to surface protein prediction, we feel that the multi-branch deep neural network (MB-DNN) approach studied in this paper may have wider appeal: Multi-omic single cell data is becoming increasingly common, and the ability to learn generalizable prediction functions between data modalities enables new integrative analyses.

In the revision we have added new analyses to illustrate how imputed immunophenotypes can improve scRNA-seq analyses. On the human cell atlas immune data, we show that the imputed immunophenotypes allow easy delineation of T and B cell subtypes. We further apply cTP-net to an acute myeloid lymphoma dataset, where imputed immunophenotypes reveal the differentiation stages of malignant cells, a step that is essential for target therapy. With the incorporation of new CITE-seq training data, cTP-net can now predict 24 different type of surface proteins, including markers for cell subtypes, such as CD45RA/CD45RO (naïve/differentiated T cells), CD27 (memory B cells), etc. With the accumulation of more training data sets, cTP-net will be able to accommodate more predictable protein targets and improve in generalization accuracy. We hope that these new analyses, and extensive rewriting, allow this paper to reach a broad audience.

Major concerns:

1. The authors describe an exemplary application of cTP-net using the HCA CBMC and BMMC data. However, the analysis highlighting CD56+/- NK subpopulations seems very "cherry picked" and is not convincing. The authors describe the visual expression pattern of a single protein in a tSNE plot. A systematic analysis is required to demonstrate that cTP-net adds meaningful value to biological analysis/interpretation, ideally over multiple data sets/scenarios. I.e. do these two subpopulations form separate clusters when including the imputed protein expression as a feature? Is the "opposing gradient" statistically significant? Across all donors? Does differential expression analysis on the imputed protein values reveal the right markers?

Thank you for this suggestion. The substratification within NK cells is previously known, but hard to identify using RNA expression levels alone (see original CITE-seq paper). Here, we illustrate that it can be apparent with the imputation of the right protein markers. To tighten the argument, we followed your suggestions and conducted a systematic analysis on the imputed CD16 vs CD56 in NK cells. We added a contour plot as Figure 3c, which clearly shows two clusters. We also computed the correlation between CD16 and CD56 and further fit a linear model (F-test p-value =1.667e-15), confirming that the high

negative correlation between CD16 and CD56 that are not due to chance. We also shown such gradient across other donors in Figure S8, S9, and S10.

2. From a deep learning perspective it is not clear why the denoising step is required. At first I thought it was required to reduce the dimensionality of the input space but (as far as I understood) the denoised matrix has the same dimensions as the original gene expression matrix. It find it hard to believe that training the network directly on the original gene expression input will not achieve equal performance (or better). Table S5 contains the performance summary of a number of different models. However, direct comparison with and without the SAVER-X denoising is not provided.

This is a good question and we have also explored this. Directly training a deep neural network for the prediction task, without prior denoising, simply doesn't perform as well, especially on generalization tasks. In the revision, we have added direct comparisons in Table S5. Without denoising, cTP-net with similar structure performed worse in validation set (0.02 decrease in correlation, which is large compared to all other parameter tweaks). We believe that denoising helps for the following reasons:

(1) The denoising method, SAVER-X, relies on an explicit and tested noise model to remove noise from the raw data. Thus, the initial denoising step does incorporate more information: The noise model and the maximum-likelihood estimated gene-specific dispersion parameters. SAVER-X makes use of the noise model to obtain estimates of the true RNA counts, and this is important because the relationship we are striving to recover is between true RNA counts and protein level. If one were to start with raw data, we would be learning a map between raw counts and protein level, which varies more across data sets and thus does not generalize well.

(2) By denoising the data, the input for learning the RNA-protein relationship is less sparse. Manifold learning on a more continuous input space usually works better.

(3) On top of the underlying autoencoder, SAVER-X performs Bayesian shrinkage between the autoencoder output and the observed data. This shrinkage step is key in preventing over-imputation (over-smoothing). See SAVER-X manuscript Figure 3 for more details.

(4) Technically, although the network obtained by merging the autoencoder in SAVER-X and cTP-net has the same expressivity, optimizing such a network is different from separately optimizing the autoencoder with the SAVER-X loss, followed by SAVER-X empirical Bayes shrinkage, followed by the training of the cTP-net prediction layers. For the above reasons, we found the latter to be better, which also provide denoised scRNA-seq data for downstream analysis.

We also added a paragraph in the discussion session to address this concern to the readers.

3. The authors should consider training cTP-net on the latent space of SAVER-X. Could this improve the model by decreasing dimensionality?

As mentioned above, the output of SAVER-X is not simply a function of this latent space. There is a Bayesian shrinkage step in SAVER-X that recovers the erratic stochasticity in single-cell level gene expression by taking a weighted average of the autoencoder output and the observed counts. See points (3) and (4) in the response to comment 2 and details in the SAVER-X manuscript. Intuitively,

there is a lot of stochasticity in RNA expression that is not captured by the latent factors (some of this can be called biological “noise”). This stochasticity propagates to protein levels, and training cTP-net on the latent space of SAVER-X would lose this information.

4. Throughout the manuscript accuracies and correlation coefficients for different models/approaches are missing standard deviations/errors. As the differences are small for some comparisons it is important for the reader to get an understanding of the uncertainties derived from running the model multiple times.

Thanks for pointing this out. We address this point by running the model multiple times starting from different random seeds. From this we computed the standard error for each evaluation statistic and incorporated them in Table S5.

5. How does cTP-net deal with over-imputation? What happens if my input is a completely unrelated cell type? What protein predictions will I get? Can the authors provide any guidance on this potential issue for users?

This point has been briefly discussed in the leave-one-cell-type analysis. For each of the high-level cell types in each data set in Table S2, all cells of the given type are held out during training, and cTP-net, trained on the rest of the cells, was then used to impute protein abundances for the held out cells. As shown in the result, such prediction still achieves higher correlation in many cases, and in all cases gives a better proxy for the level of the surface protein than the corresponding raw RNA count. However, one should not impute on a cell type that is very different from all cell types in the training set. For example, if we trained only on T cells and predict on monocytes, we shouldn't expect a satisfying result. We added some text in the discussion to address this nuanced issue.

6. As the authors correctly point out in their SAVER-X manuscript different scRNAseq technologies differ in the underlying model assumptions (NB vs ZINB). Can cTP-net successfully predict across droplet and well based scRNAseq technologies?

Yes, cTP-net, without any model assumptions, should do well across all UMI-based scRNA-seq datasets as mentioned in the CITE-seq paper. All the datasets we applied are sequenced by Drop-seq and 10X, but may not work on data without UMIs. We have included this point in the discussion section.

7. I was intrigued by the interpolation analysis but would like to see more analyses towards understanding the properties of the network. I.e. if data was clustered using the activations in the 128 neuron layer, would this return the various cell types/protein expression clusters?

This is indeed an interesting point. Given we are predicting the protein abundance directly from the bottleneck layer, it should capture the cell stage and differentiation variability. With our additional analysis in the interpolation analysis section and Figure S6, we show that such bottleneck recapitulate clusters indicating both cell types and protein expression.

Minor concerns:

1. *Additional information with respect to training is missing. I.e. number of epochs, early stopping etc?*

Thank you for this suggestion. We have added more training details in the method section.

2. *p. 3 typo – "... these out-of-cell-type predictions are, as expected, are still greatly improve upon the corresponding RNAs while slightly inferior in accuracy to the traditional holdout validation ..."*

Thanks for pointing this out. The typo is fixed.

Reviewer #2 (Remarks to the Author):

In this paper the authors introduce cTP-net (single cell Transcriptome to Protein prediction with deep neural network), a deep learning tool which leverages single-cell sequencing data to predict cell-surface protein data. The tool shows promising results, especially with regards to its accuracy when translating to novel cell types. Furthermore, as multi-omic training data continue to become more prevalent, the model will continue to improve in its precision and hence applicability. This being said, I feel that the paper needs to be significantly improved before it can be published. In particular, the methodology should be better motivated from a biological point of view. The model should be evaluated against cell-type prediction model since proteins are often used to predict cell-types. The denoising part needs to be better motivated.

Thank you for taking the time to review our paper. Yes, we agree with your sentiments that the motivation for predicting surface proteins should be explained better. In the revision, we've added two new analyses where the motivation is made clear:

1. On the human cell atlas data set, the imputed levels of canonical surface protein markers are used to confidently delineate T and B cell subtypes. Due to their belonging to a continuous trajectory, some of these subtypes are tricky to identify by RNA expression. Here, the imputation allows us to orient the cells under the classical immunophenotyping framework, where the imputed gradients allow easy visualization and confident cell state assignment.
2. On a recently published acute myeloid leukemia data set, the imputed levels of myeloid differentiation markers are used to identify the cell-of-origin differentiation stage of malignant cells. This is incredibly complicated to do using the RNA expression counts, because the transition to malignancy has already grossly changed the transcriptional profile of the cell. The original study sought to do this by aligning the cells onto a differentiation trajectory obtained from healthy cells, which masks malignancy-specific signatures.

We hope that these newly added analyses give the reader a better sense of the motivation of the method and how it can be applied.

On comparisons to cell-type prediction models: We want to clarify that the objective for cTP-net is to predict the relative surface protein abundance, which is useful for cell type labeling and characterization

and, sometimes, for the discovery of novel cell types. Since much of what we know about immune cells is through their surface marker composition, and since surface proteins are key for drug targeting and for spatiotemporal cell imaging, the prediction of surface proteins on scRNA-seq data is important, per se. If one wishes to perform de novo cell type identification using the imputed protein markers, one would still need to rely on the existing clustering tools.

Also, we want to mention that, by incorporating new CITE-seq data since the first submission, cTP-net can now predict 24 different type of surface proteins, including markers for cell subtypes, such as CD45RA/CD45RO (naïve/differentiated T cells), CD27 (memory B cells), etc. With the accumulation of more training data sets, cTP-net will be able to accommodate more predictable protein targets and improve in generalization accuracy.

Major comments.

1. - The organization of the paper is slightly scattered. More sections and headers would help. Specifically the paper lacks an introduction section and a discussion section.

Thanks for pointing this out. We have added an introduction section and a discussion section. We also divided the results into sub-sections to help the reader navigate the manuscript.

2. - The paper lacks discussion of how the model predictions can be used in a broader context. Perhaps presenting an example analysis workflow where this method is integral would make its usefulness more clear. It's not clear what one would do with the predicted protein expression. It would be good to present a motivating example. More specifically, it seems that the main application here is in predicting cell type. In this case, one wonders whether it is really necessary to predict cell types.

We agree. As described above, we have added two additional case studies: A more in depth analysis of B and T cell subtypes on the HCA data set, and a new analysis of a recent acute myeloid leukemia data set. Please see the response to your introductory comments above. We hope these make the motivation and biological interpretations more clear.

3. - It's not clear to me what one needs to perform denoising before prediction. One would expect the model to be able to learn this as part of the process. So more comparisons need to be made with and without imputation. And why SAVER-X? Does it matter, could you use another tool?

This is a good question and we have also explored this. Directly training a deep neural network for the prediction task, without prior denoising, simply doesn't perform as well, especially on generalization tasks. In the revision, we have added direct comparisons in Table S5. Without denoising, cTP-net with similar structure performed worse (0.02 decrease in correlation, which is large compared to all other tweaks one can do). We believe that denoising helps for the following reasons:

(1) The denoising method, SAVER-X, relies on an explicit and tested noise model to remove noise from the raw data. Thus, the initial denoising step does incorporate more information: The noise model

and the maximum-likelihood estimated gene-specific dispersion parameters. SAVER-X makes use of the noise model to obtain estimates of the true RNA counts, and this is important because the relationship we are striving to recover is between true RNA counts and protein level. If one were to start with raw data, we would be learning a map between raw counts and protein level, which varies more across data sets and thus does not generalize well.

(2) By denoising the data, the input for learning the RNA-protein relationship is less sparse. Manifold learning on a more continuous input space usually works better.

(3) On top of the underlying autoencoder, SAVER-X performs Bayesian shrinkage between the autoencoder output and the observed data. This shrinkage step is key in preventing over-imputation (over-smoothing). See SAVER-X manuscript Figure 3 for more details.

(4) Technically, although the network obtained by merging the autoencoder in SAVER-X and cTP-Net has the same expressivity, optimizing such a network is different from separately optimizing the autoencoder with the SAVER-X loss, followed by SAVER-X empirical Bayes shrinkage, followed by the training of the cTP-net prediction layers. For the above reasons, we found the latter to be better.

We have not tried it with another denoising tool. In our experience SAVER-X has the highest accuracy (see our comparisons in <https://www.nature.com/articles/s41592-019-0537-1> and a third-party benchmark in <https://f1000research.com/articles/7-1740>), and thus we cannot guarantee that the results would hold if another denoising tool is used in cTP-net training procedure.

We also added a paragraph in the discussion section to address this concern to the readers.

4. - 109-119. It would be nice to explicitly separate the two ways described. From what I understand, there are two steps a model takes when computing protein expression: (1) identify cell-type markers to identify cell type, which helps estimate protein counts, and/or (2) use GEX patterns to directly identify protein expression. I think the argument in the paper is that Seurat v3 fails to do the second step, whereas cTP-net does both. Then this benefit of cTP-net would give it an extra edge over Seurat particularly when it comes to variations that are not explained through cell-type differentiation.

This is a nice way to clarify. However, instead of two steps, we think it's more apt to consider cTP-net as using two levels of information: Variation in protein expression can be decomposed into (1) cell type level protein expression mean and (2) within cell type variation (gradient). Seurat v3 can recover the former but is not very accurate in recovering the latter (as seen from Figure 2 b,c). But, as mentioned in the first paragraph of "Comparison to Seurat v3" section, this is not the only distinction between Seurat v3 and cTP-net. cTP-net directly learns a mapping function from RNA to protein. If the function generalizes well, then we can predict on unseen cell types (as we demonstrated through the leave-one-cell-type-out analysis). Seurat v3, on the other hand, is based on the Mutual Nearest Neighbor algorithm, which is not very accurate in extrapolation.

5. - The paper does not explicitly show the model's performance compared to purely using cell-type marker genes. It is not clear just how much of the performance can be easily explained by good use of cell-type marker genes. It would be good to use the approach with and without marker genes. For example, when predicting CD3 and CD4, the respective genes should be excluded.

We interpret this comment as asking: How much does the prediction of the protein levels depend simply on cell type? We agree that this is a valid and interesting question. In the paper, we have tried our best to address the predictive role of cell type through Section “ Network interpretation and feature importance”, where we performed an interpolation analysis. In short, this analysis calculate, for each gene, how much does its RNA expression level influence the expression of a protein. Then, pathway analysis can be performed on the high ranking genes. Through such an analysis, we did find that high ranking genes make good markers for cell type, but there are also many other genes that are influential, see Table S3.

We’ve also thought of doing predictions after removal of “cell type markers”. The difficulty here is that cell type markers for most cell types are not well defined, and there is a high level of redundancy. Apart from the traditional well known markers, there are many other genes which are highly informative of cell type. Furthermore, cell type, itself, is vague concept. Part of the reason for this analysis is to characterize intra-celltype heterogeneity through the combined use of RNA and imputed protein levels.

6. - 129-130. I don't understand how this demonstrates that cTP-net uses both types of gene in its predictions. From what I understand, you are again saying that cTP-net can use GEX patterns for more than just determining cell type, but also to directly predict protein expression. To me this point is just a restatement of the previous paragraph, so I think that the connection between the two points could be made more clear.

Thanks for pointing out this confusion. The previous paragraph illustrated that protein levels imputed by cTP-net reveals within cell type gradients. With the interpolation analysis, we simply want to answer the question: Which are the genes that have the highest impact in the prediction accuracy. This result is: Cell type markers in the population level; RNA processing, RNA binding, protein localization, biosynthetic process related gene in the within cell type level. We rewrote this section to clarify. Hopefully it is no longer confusing!

7. – 140. Related to the points above many tools have already been published to predict cell type from gene expression. So your proposed approach should be compared to these tools when the end goal is cell type.

Thank you for the suggestion. However, as we discussed above, this paper is not about cell type identification. The objective of this manuscript is to develop a method that can predict surface protein abundances. The abundance of canonical surface protein markers can help the cell type labeling process, but this does not need to be the end goal. Please see also our reply to your introductory comments above.

8. – 281-292. Why use these particular settings? Can you provide an intuitive explanation? See comment for table S5. I think it would be good to perform a better sensitivity analysis to see how the parameters affect performance.

Thank you for the comments. We included in the revision a more thorough write-up of our analysis of how parameter settings affect performance. Please see the comments on Table S5 below.

9. - 318-329. *I find this section very confusing. Specifically, I cannot understand the line 'for genes within *gs*, cell labels were permuted'. However, this whole section could use more/clearer explanation.*

Thanks for pointing out this confusion. We have rewritten this paragraph. We hope that the interpolation analysis procedure is clear now.

Minor comments:

- *Fig 2. The aesthetics are cluttered. It might be better to leave a part out or break this into separate figures.*

Thanks for the comment. We have broken this figure into two (now Figures 2 and 3).

- *S1. It is not clear how the correlation is calculated.*

We have edited this figure to make it more straightforward.

- *S8/S9. It might be nice to also show the actual counts for each protein, or at least briefly address the lack thereof. I am curious if there are cases where cTP-net generated protein values differ significantly from the actual counts.*

Sorry for the confusion. S8/S9 consist of cells from the Human Cell Atlas data set, where no actual protein count was measured. This serve as an example to show how imputed protein abundances support previous knowledge based on cell types.

- *Table S5. Why were these particular changes chosen? I think few people would argue that the MB structure is bad, but perhaps other changes would make a difference. Here are other adaptations I might have liked to see tested, in order of importance:*

- *dropout layer (probably before the first hidden layer)*
- *activation functions (they only use relu and don't consider any others)*
- *number of layers (this would help model nonlinear parts of the function)*
- *learning rate:*
- *number of nodes (what about reducing the bottleneck?)*
- *others? (loss function /optimization function /preprocessing)*

We have now tested multiple different types of models listed in Table S5 of the revision. We have actually tried many of these model before, but for the sake of brevity, we didn't include these in the original submission. We still show that the current the model is the best as measured by test accuracy.

- *Figure 2A, why is the correlation for CD16 so weak?*

CD16 is generally harder to predict in PBMC data. However, in CBMC, it is much more predictable as shown in the Supplement Figure 4. This is due to the fact that there are few NK cells in the original

PBMC training dataset, as compared to the CBMC data set, and CD16 has zero expression in other cell types (except a small handful of CD16+ monocytes).

- *Figure 2C is unclear, what are we looking at? What does CD19, DC means? CD19 is not expressed in DCs as it is a B-cell surface marker.*

Thank you for the pointing this out. This is the within cell-type prediction accuracy comparison between cTP-net and Seurat v3. We have modified the figure for easier interpretation. Yes, CD19 in DC is not that meaningful, and in the revision we have selected several more biological meaningful protein-cell type pair for comparison.

Minor fixes:

- *30. become → becomes*
- *61. CBMC's are cord blood mononuclear cells (not cordical)*
- *77. grammar*
- *80-81. it seems misleading to keep referencing this proxy as though people expect it to be highly accurate.*
- *290. "where the loss [is] l1 norm"*
- *S8/S9 legend → nature killer cells should be natural killer cells*

These are now fixed. Thanks for spotting them.

Minorer fixes:

- *grammar on 69/70 is awkward*
- *289. I am not familiar with the notation (the 1's on the right)*
- *S9 → NK label missing in BM2 figure*

These are also fixed.

REVIEWERS' COMMENTS:

Reviewer #1 (Remarks to the Author):

The authors have successfully addressed all my major concerns.

Here are some typos:

P4, 95 – “We did this for each cell type and generated an ‘out-of-cell-type’ prediction for every CELLS.”

P5, 113 – “...we trained a network using THE all cell populations combined...”

Reviewer #2 (Remarks to the Author):

While I think that the paper has improved. I still think that the authors fail to demonstrate the value of protein expression besides cell labeling. The new example on the HCA data is not terribly convincing. The authors say “Due to their belonging to a continuous trajectory, some of these subtypes are tricky to identify by RNA expression.” Can the authors elaborate on this and perhaps quantify this?

Also the authors haven't really responded to one on my original queries, namely:

“The paper does not explicitly show the model's performance compared to purely using cell-type marker genes. It is not clear just how much of the performance can be easily explained by good use of cell-type marker genes. It would be good to use the approach with and without marker genes. For example, when predicting CD3 and CD4, the respective genes should be excluded.”
When predicting specific proteins, it would be interesting to remove the corresponding genes from the RNA-seq data to see how well the model still predicts.

REVIEWERS' COMMENTS:

Reviewer #1 (Remarks to the Author):

The authors have successfully addressed all my major concerns.

Here are some typos:

P4, 95 – “We did this for each cell type and generated an ‘out-of-cell-type’ prediction for every CELLS.”

P5, 113 – “...we trained a network using THE all cell populations combined...”

Thanks for pointing these out. The typos are fixed.

Reviewer #2 (Remarks to the Author):

While I think that the paper has improved. I still think that the authors fail to demonstrate the value of protein expression besides cell labeling. The new example on the HCA data is not terribly convincing. The authors say “Due to their belonging to a continuous trajectory, some of these subtypes are tricky to identify by RNA expression.” Can the authors elaborate on this and perhaps quantify this?

We thank the reviewer for the comments. Immune subtypes are not well separated in low dimensional representations, and don't belong to discrete clusters. In fact, in some cases, for example T cell activation, the underlying biology manifests as a continuous scale of phenotypes. This is what we mean by “tricky”. Indeed, in this paper we focus heavily on the cell type labeling, but we believe that is one of the main challenges in single cell transcriptomics today, and addressing this problem in the immune context if, we feel, an important contribution.

Also the authors haven't really responded to one on my original queries, namely: “The paper does not explicitly show the model's performance compared to purely using cell-type marker genes. It is not clear just how much of the performance can be easily explained by good use of cell-type marker genes. It would be good to use the approach with and without marker genes. For example, when predicting CD3 and CD4, the respective genes should be excluded.” When predicting specific proteins, it would be interesting to remove the corresponding genes from the RNA-seq data to see how well the model still predicts.

We apologize for missing this point in the previous revision. We have trained a model without the corresponding genes of all the predicted surface proteins. The result shows almost similar performance (correlation 0.967), which is due to (1) redundant of cell surface marker genes and (2) predictability of these genes by other genes. We have added sentences at page 7 to address this point.